# Psychosocial work factors and sick leave risk after a terrorist bomb attack: a survey and registry-based longitudinal study of governmental employees in Norway

Maria Teresa Grønning Dale [1,2] Alexander Nissen [1] Mona Berthelsen,[1] Håkon Kristian Gjessing,[3,4] Trond Heir[1,5]

¹Norwegian Center for Violence and Traumatic Stress Studies, Oslo, Norway
²Department of Psychology, University of Oslo, Oslo, Norway
³Centre for Fertility and Health, Norwegian Institute of Public Health, Oslo, Norway
⁴Department of Global Public Health and Primary Care, Faculty of Medicine, University of Bergen, Bergen, Norway
⁵Institute of Clinical Medicine, Faculty of Medicine, University of Oslo, Oslo, Norway

**Correspondence to**
Dr Maria Teresa Grønning Dale;
m.t.g.dale@nkvts.no

## ABSTRACT

**Objectives** Studies show that social support may reduce the negative psychological effects of terror. The aim was to explore the effects of the psychosocial work environment on sick leave risk among governmental employees after a workplace bomb attack.

**Design** We linked longitudinal survey data collected at 10 and 22 months after the bombing with registry data on doctor-certified sick leave collected from 42 months before the attack to 33 months after the attack. ORs and rate ratios were estimated with mixed effects hurdle models.

**Setting** The bombing of the government ministries in Oslo, Norway, 22 July 2011.

**Participants** We identified 1625 participants from a cohort of 3520 employees working in the ministries during the bombing in 2011.

**Results** After adjustment for confounders, social support from coworkers reduced the odds of sick leave (OR 0.80, 95% CI 0.68 to 0.93), and there was marginal evidence for reduced odds with support from superior (OR 0.87, 95% CI 0.87 to 1.03). A social work climate, an innovative climate and a human resource primacy climate (HRP) reduced the sick leave risk (eg, HRP OR 0.77, 95% CI 0.66 to 0.90). The hurdle model found no associations between psychosocial support at work and the duration of sick leave.

**Conclusions** Psychosocial support at work can enhance employees' work ability after terror and reduce the sick leave risk by more than 20%. However, a supportive psychosocial work environment did not reduce the duration of sickness absence. The protective role of psychosocial work factors on sick leave may be most significant when employees are at work and interact with their work environment.

## INTRODUCTION

When terrorism strikes the workplace, where people spend much of their time, survivors are highly affected.[1] The workplace serves as an essentially social context that provides routines, purpose, economic and social resources to one's life, all of which can be impaired after workplace violence.[2] After the Oslo bombing in 2011, terror-exposed individuals were at high risk of post-traumatic stress disorder (PTSD), depression and increased sick leave.[3 4] The magnitude of terror-related stressors might have a different impact on survivors. Factors shown to affect the sickness absence are health, age, gender, coping style, personality, physical and psychosocial work factors, work schedule characteristics and available social resources.[5–7] A common assumption is that the cumulative effects of negative life events can cause psychosocial morbidity, which is highly relevant after major disasters considering the many challenges that arise in the aftermath of such events.[8 9] Previous research has often focused on post-traumatic stress reactions and severity of psychopathology. However, many trauma-exposed individuals display high levels of

resilience and coping in the wake of disaster, which is also apparent in prior studies by our research group. Specifically, a high proportion of government employees had a strong sense of attachment and commitment to the workplace, and some managed to work despite a very high symptom load.[3] [10] In particular, it seems like the social network at work combined with a supportive organisational climate can be a significant source of support for employees.

### Strengths and limitations of this study

► This longitudinal study combines survey and registry data and examines the sick leave risk in a large population exposed to a workplace terror attack.
► This study has a relatively high response rate and an appropriate number of participants with and without sick leave, making it possible to compare the two samples.
► In terms of generalisability, the study sample consisted of a majority of highly educated government officials and bureaucrats.
► The use of self-reported data on psychosocial work environments may not objectively reflect available social support.
► A comprehensive approach that incorporates both intrinsic and extrinsic factors is needed to understand better how individuals cope in the aftermath of terror.

Social support and good social relations affect health and act as protective resources against the negative impact of major life events.[11] [12] Social support involves instrumental (eg, material and financial), informational (eg, advice and guidance) and emotional (eg, empathy, trust and emotional venting) resources.[13] Previous studies show that these three supportive components can be directly associated with overall lower levels of psychological distress, independent of exposure to trauma.[14] Additionally, social support can act as a buffer by reducing or taking away the impact of major trauma.[15] In line with this, studies show that after a natural disaster, low levels of social support have been associated with higher levels of PTSD and depression.[16–18] Further, when experiencing intense stress from terror, such as the September 2011 terrorist attacks, research indicates that turning to others for help may prevent the development of long-lasting psychological sequelae.[19] Although the social network at work is considered to be more formal than social relations to family and friends, research shows a strong association between support from colleagues and superiors and reduced sick leave.[20–22] However, the vast majority of studies on the effect of the psychosocial work environment on sick leave risk have not focused on workers exposed to a terror attack.

At present, it is unclear how the psychosocial work environment influences the sick leave risk and the duration of sick leave after terror. We assume that a high degree of social support from leaders and colleagues, combined with a supportive organisational climate with a concern for human resources, are essential for employees struggling with returning to work after unexpected and threatening workplace violence. Our study sample shared an extraordinary experience, and psychosocial support at work may be even more critical for re-establishing routines and recovery of trust and safety. The present study aims to explore whether increasing levels of psychosocial support are associated with a corresponding decrease in employees' sick leave risk and reduction in sick leave days and consequently add knowledge to the limited literature on this association.

## METHODS
### Study population and data sources
This prospective cohort study includes web-based survey-data combined with registry data on doctor certified sick leave from ministerial employees in 14 of 17 ministries, after a car bomb attack at the Norwegian government

offices in Oslo, 22 July 2011. The terror bombing caused substantial damage to buildings and infrastructure, killing eight and injuring 209 people. Negative health reactions were prevalent among all employees, and the present study includes all, whether indirectly or directly, exposed to the bomb. 10.5% of the employees were classified as directly exposed as they reported to be in the government district during the bomb explosion.[4]

The Norwegian Centre for Violence and Traumatic Stress Studies conducted the survey in collaboration with the National Institute of Occupational Health in Norway 10 and 22 months after the terrorist attack. Eligible participants were informed about the study through their ministries and received an invitation letter containing a unique log-in code to access the Web-based survey, including information on withdrawal procedures. Data on doctor certified sick leave was obtained from Statistics Norway and the Norwegian Labour and Welfare Administration.

For the purpose of this study, we used survey data on employees' background variables and information on the psychosocial work environment 10 months after the attack (T1) and 22 months after the attack (T2). The survey data were linked with registry data on doctor certified sick leave from a period of 42 months before the attack to 33 months after the attack. However, as the survey data was collected over a period of 4 months (about 8–12 months and 20–22 after the attack), we used registry data on sick leave from 13 to 21 months after the first survey (T1) and from 25 to 33 months after the second wave of the survey (T2) (see figure 1).

All employees provided informed consent, and strict procedures were followed to ensure confidentiality. Willing participants received a postal invitation letter containing information about the study and withdrawal procedures. In the study's invitation letter, each employee was assigned a unique project identification ID-number and a log in code to access the study's Web-based questionnaire. Once the participants logged in to the Web-based questionnaire, they were informed that filling out the questionnaire was equivalent to written consent of participation in the study. Further, they had to thick off 'yes' to the question of linking survey data to registry data on doctor certified sick leave. Based on the personal identification number from the Norwegian Population Register, Statistics Norway performed data linkage and deidentification.

Invited participants were ministerial employees who were employed in the Norwegian ministries at the time of the attack. Three of the initial 17 ministries did not agree to the study's consent procedure (two ministries; n=440), or the office was located approximately 1 km away from the government district with a significant proportion of the workforce based abroad (one ministry; n=856). 3520 invited employees consented to participate; 59 could not be reached with information about the study, and 482 employees left the ministerial job or changed ministry affiliation prior to study completion. The survey response rate was 56% (1956 of 3520), where 1023 employees did

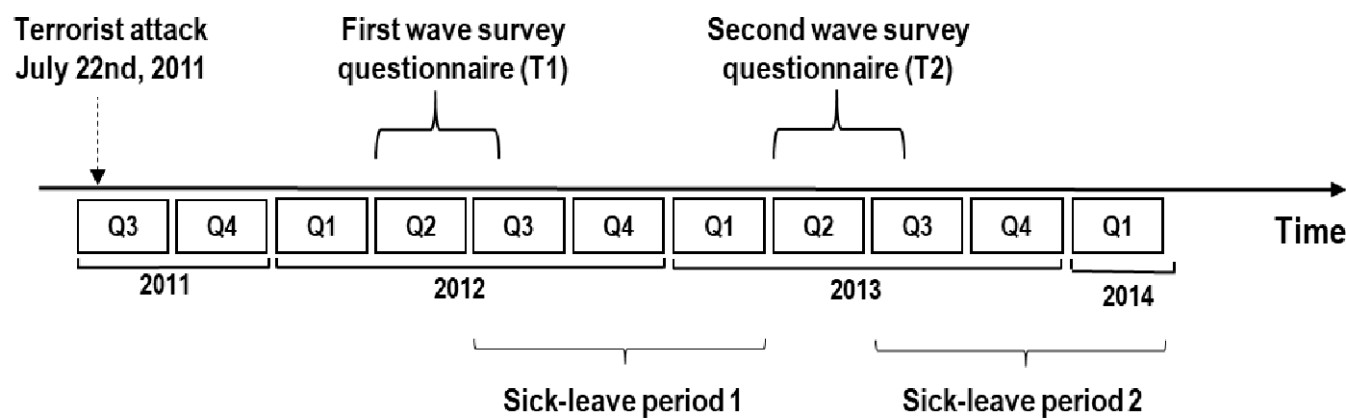

**Q:** Quarter
**T1:** Starts about 10 months after the attack
**T2:** Starts about 22 months after the attack

**Figure 1** Timeline for sick leave outcomes and survey measures on psychosocial work environment.

not participate in the survey at any time point (at T1 and T2), while 1061 participated at both T1 and T2. In this study population, we identified 1625 participants eligible for the study, all with relevant data on the psychosocial work environment at T1 and/or T2 and with registry data for the 9 months following T1 and T2. Further details on design and participants have previously been described in a recent article from our research group.[3] See figure 2 for more information on participants eligible for this study.

### Patient and public involvement

There is no direct patient involvement in this study. Neither patients nor the public was directly involved in our research's design, conduct, reporting or dissemination plans.

### Psychosocial work exposures

Social support at work (comprising two subscales) and supportive organisational culture (comprising three subscales) were measured by the General Nordic Questionnaire for Psychological and Social Factors at Work, QPSNordic.[23] All responses were scored on an ordinal five-point scale ranging from '1=very seldom or never' to' 5=very often or always', and missing response to one of the items comprising each scale was allowed when computing a mean sum score.

Social support from coworkers was measured with two items, and a typical item was 'If needed, can you get support and help with your work from your coworkers?' Social support from superior comprised three items. A typical item was, 'If needed, is your immediate superior willing to listen to your work-related problems?' Three subscales measured organisational culture: social climate, human resource primacy climate (HRP), and innovative climate, all comprising three items. An item representing social organisational climate was 'Is the climate

encouraging and supportive in your work unit?' A typical item measuring HRP was 'Are workers well taken care of in your organisation?' To measure innovative climate, a relevant item was 'Are workers encouraged to think of ways to do things better at your workplace?' Reliability tests of the five subscales have demonstrated approvable internal consistency measured with Chronbach's alpha from 0.71 to 0.83 and test–retest reliability from 0.72 to 0.83 with more than 5 weeks interval.[24]

### Sick leave

The outcome of this study, doctor-certified sickness absence, was based on registry data on employment from Statistics Norway and registry data on sick leave from the Norwegian Labour and Welfare Administration. The former registry contains the number of expected workdays per quarter for a given person based on the person's contract(s) of employment. Weekends, public holidays and days of vacation are not considered potential workdays. For a person with full-time employment, there are roughly 170 expected workdays in each 9-month period explored in the study. The registry on sick leave contains the number of days a given person was absent from work per quarter due to doctor-certified sick leave. The registry takes account of whether the person works full time or part time, and whether sick leave was graded or not (eg, a person may be on 50% sick leave). For example, if a person with 80% employment (ie, four expected workdays per week) gets 1 week of 50% doctor-certified sick leave in a quarter, the person will have two registered days of sickness absence for that quarter. Sick leave prior to the attack (from the first quarter of 2008 until the second quarter of 2011) was included as a potential confounder and defined as total sickness absence days divided by the number of expected workdays registered for this period.

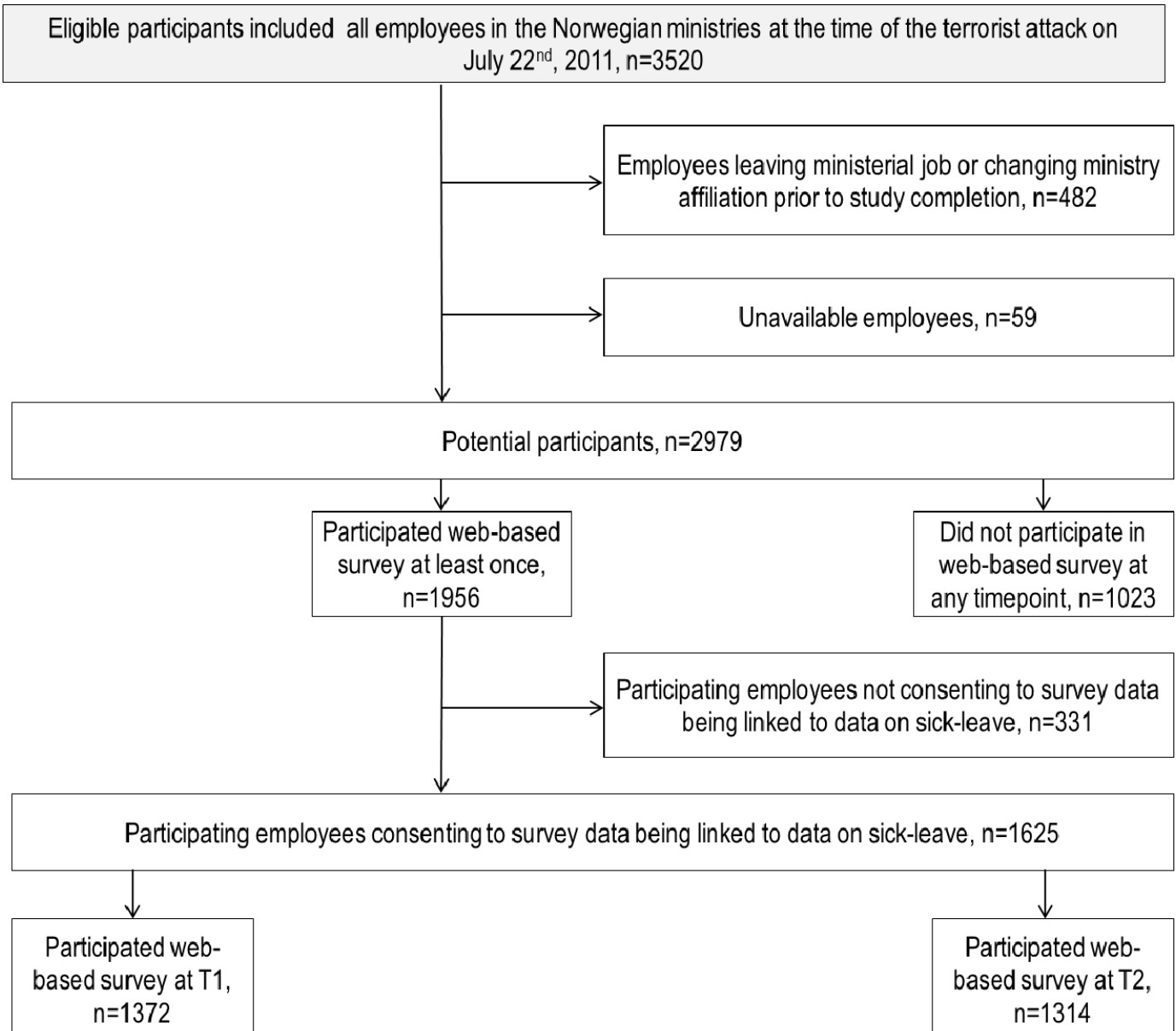

**Figure 2** Flow chart displaying participant disposition.

When used as an outcome variable in the main analysis, sick leave was examined in the two 9-month periods following the survey points at T1 and T2 (see figure 1 for time frames).

### Covariates

Covariates that could potentially influence the association between the psychosocial work environment and sick leave risk were considered a priori as potential confounders. The following covariates were included in the main analysis: time measured by the two survey waves (T1 contrasted with T2), sex, age, education, direct versus indirect exposure to terror and sick leave prior to the attack. Proximity to the bomb explosion was assessed by asking employees where they were located during the explosion. Participants were given five alternative responses: (1) in the government district; (2) in

downtown Oslo, but not in the government district; (3) in Oslo, but not downtown; (4) in Norway, but not in Oslo and (5) abroad. Only employees in the government district when the bomb exploded were defined as directly exposed to terror.

### Analytical strategy

Mixed effects hurdle models were used in analyses, and participants contributed with data as long as they had at least one time point without missing values. As a result, the participants were included in the analysis as long as they had answered all variables used in the model for at least one survey wave (T1 and/or T2).

The frequency of sick leave was expressed as count data, with an excess of zeros (no sick leave).[25] As the count data was overdispersed with variance larger than the mean, we used negative binomial hurdle models to estimate

the incidence rate of sick leave.[26 27] Hurdle models are two-part models. The first part uses a logistic regression model to estimate the odds of the outcome being above zero vs zero for various predictor levels in the model, summarised as ORs compared with a set reference. In this study, we estimated the OR of sick leave among employees according to various psychosocial work environments, using crude (OR) and adjusted (aOR) ORs with 95% CIs. The ORs compare the odds of having at least 1 day of sickness absence versus no sickness absence for various predictor levels. The second part of the hurdle model studies the mean number of days of sickness absence, conditional on having at least 1 day of absence. For this part, it uses a truncated negative binomial regression model, where zero has been excluded for the positive counts.[28] By fitting the negative binomial model, we obtain mean count ratios, or rate ratios (RRs, with 95% CIs). The RR compares the mean number of days of sickness absence over levels of various predictors, including the main exposure. Consequently, the RRs estimate the exposure effect on the number of sickness absence days among those with at least 1 day of absence.

To account for varying person-time at risk (ie, not all employees worked full-time during the observation period), the hurdle models were offset by employees' expected workdays in the relevant periods. Since data were collected longitudinally, with repeated measurements on individuals, we used a mixed effects extension of the hurdle model, with a random intercept on individuals. The effects of social work factors were modelled in five separate hurdle regression analyses. Each work factor was included as a continuous variable, with raw mean scores, in the model.

The binomial regression analyses and the zero-truncated negative binominal analysis were computed in STATA V.15 (STATA) and R, using the R package GLMMadaptive.

## RESULTS
### Characteristics of the study population
In the study population of 1625 employees, the prevalence of sick leave from 13 to 21 months after the attack was 25% (421 of 1626) after excluding employees leaving their job or changing ministry affiliation prior to study completion. After the second survey wave (T2), in the period from 25 months to 33 months after the attack, the sick leave prevalence was 24% (394 of 1625). Further, the mean number of sickness absence days was about 25 days at 13 to 21 months after the first survey (T1) and decreased to 22 days in the period of 25 to 33 months after the second wave of the survey (T2) among employees with sick leave. When expressed as the overall percentage among all employees in this study, this corresponds to 4.3% and 3.8% sick leave of all working days.

Table 1 shows the distribution of background variables for sex, age, educational level, relationship status, directly and indirectly exposed, psychosocial work environment and sick leave, all retrieved from the questionnaire survey completed 10 months (T1) and 22 months (T2) after the terror attack.

Table 1 compares participants with and without sick leave 10 months after the attack (T1) and 22 months after the attack (T2). At both measurement points (T1 and T2), the group with sick leave had overall lower levels of education (p<0.01) and a higher percentage of females (71%) compared with the group without sick leave. At T1 there were more participants directly exposed to the bomb explosion among those with sick leave (14%) than those without sick leave (10%). Further, at both measurement points (T1 and T2), the group with sick leave reported overall lower scores in social support from coworkers (p<0.01), social organisational climate (T1: p=0.02; T2: p<0.01), and human resource primacy climate (p<0.01). At T1 the group with sick leave reported lower scores in support from leaders (p<0.01), and at T2 those with sick leave reported lower scores in innovative organisational climate (p=0.02). The groups with sick leave did not differ significantly from participants without sick leave with regard to marital status and age at T1 and T2.

### Psychosocial support at work and sick leave risk
After adjusting for confounders, we observed that high support from coworkers was associated with overall reduced odds of sick leave for all time periods (aOR 0.80, 95% CI 0.68 to 0.93). Similarly, support from superiors indicates reduced odds of sick leave (aOR 0.87, 95% CI 0.79 to 1.01), but did not reach significance in the model (p=0.063). A work environment with a social, innovative and human resource primacy climate reduced the odds of sick leave, where the estimates for all these three organisational climate variables are very similar with OR around 0.80. (eg, HRP: OR 0.77, 95% CI 0.66 to 0.90). Moreover, the hurdle model estimated the duration of sick leave (RR) among those with sick leave. We found no significant associations between social support from superiors and coworkers and the duration of sickness absence among those with sick leave, nor between a supportive HRP, social climate and innovative organisational culture and the duration of sickness absence among those with sick leave (see table 2). The RR-estimates indicate that all five factors measuring a supportive psychosocial work environment had no substantial impact on the duration of sick leave. Further, the overall effects of time (T1–T2) on sick leave scores after the attack were not significant (eg, OR 0.94, 95% CI 0.76 to 1.18 and RR 0.85, 95% CI 0.71 to 1.02), indicating unchanged odds and RR for sick leave at T2 when compared with T1 (the estimates for confounders are not shown in table 2).

## DISCUSSION
This is the first cohort study based on registry data on sick leave exploring longitudinally whether a supportive psychosocial work environment reduces the risk and duration of sickness absence after a terrorist attack on

**Table 1** Characteristics in cases with no sick leave and those with sick leave at 10 months (T1) and 22 months (T2) after the 2011 terrorist attack in the cohort with 1625 Norwegian ministerial employees

| Characteristics | T1 | | | | T2 | | | |
|---|---|---|---|---|---|---|---|---|
| | No sick leave n=1023 | Sick leave n=349 | χ²/F | P value* | No sick leave n=990 | Sick leave n=324 | χ²/F | P value* |
| Females, n (%) | 528 (51.6) | 248 (71.1)† | 40.1 | <0.01 | 525 (53.0) | 229 (70.7)† | 31.1 | <0.01 |
| Education, n (%) | | | 41.7 | | | | 38.4 | <0.01 |
| <13 years | 82 (8.0) | 68 (19.5)† | | | 92 (9.4) | 72 (22.2)† | | |
| 13–16 years | 224 (21.9) | 89 (25.5)† | | | 218 (22.1) | 73 (22.5) | | |
| >16 years | 717 (70.1) | 192 (55.0)† | | | 677 (68.5) | 179 (55.3)† | | |
| Married/cohabiting, n (%) | 598 (75.5) | 183 (68.3) | 5.4 | 0.02 | 730 (73.7) | 245 (75.9) | 0.6 | 0.45 |
| Age (m±SD) | 46.3±10.6 | 47.3±10.5 | 2.3 | 0.14 | 46.4±10.4 | 47.6±10.7 | 3.2 | 0.07 |
| Directly exposed, n (%) | 100 (9.8) | 49 (14.0)† | 4.9 | 0.03 | 112 (11.4) | 32 (9.9) | 0.5 | 0.47 |
| Sick-leave days (m±SD)‡ | | 24.8±34.6 | | | | 20.8±25.7 | | |
| Support from superior, score 1–5 (m±SD) | 4.0±0.8 | 3.8±0.9† | 11.5 | <0.01 | 4.0±0.9 | 3.9±0.8 | 3.7 | 0.06 |
| Support from coworkers, score 1–5 (m±SD) | 4.1±0.7 | 3.9±0.8† | 8.6 | <0.01 | 4.1±0.7 | 4.0±0.8† | 7.4 | <0.01 |
| Social organisational climate, score 1–5 (m±SD) | 3.9±0.7 | 3.8±0.7† | 5.4 | 0.02 | 3.9±0.7 | 3.8±0.7† | 10.2 | <0.01 |
| Human resource primacy climate score 1–5 (m±SD) | 3.4±0.7 | 3.2±0.8† | 18.5 | <0.01 | 3.4±0.7 | 3.2±0.8† | 7.3 | <0.01 |
| Innovative organisational climate score 1–5 (m±SD) | 3.6±0.7 | 3.5±0.8 | 2.9 | 0.08 | 3.6±0.7 | 3.5±0.7† | 5.4 | 0.02 |

Table showing case numbers and within-group percentages.
*P values were calculated using ANOVA for continuous variables and χ² test for categorical variables.
†Differs significantly from those without sick leave.
‡Nine-month period following the survey questionnaires at T1 and T2.
ANOVA, analysis of variance.

**Table 2** Two-part hurdle mixed effects models on work factors among 1625 Norwegian ministerial employees measured 10 and 22 months after the terrorist attack 22 July 2011, and risk of sick leave measured 9 months after each survey

| Variables | Part I: Binary logistic model | | Part II: Negative binomial count model | |
|---|---|---|---|---|
| | OR | OR* | RR | RR* |
| Psychosocial work environment | | | | |
| Social support/interactions | | | | |
| Support from superior | 0.81 (0.70 to 0.93)† | 0.87 (0.76 to 1.01) | 0.90 (0.80 to 1.00) | 0.94 (0.84 to 1.05) |
| Support from coworkers | 0.75 (0.64 to 0.89)† | 0.80 (0.68 to 0.93)† | 0.89 (0.79 to 1.01) | 0.94 (0.83 to 1.07) |
| Organisational culture | | | | |
| Social organisational climate | 0.72 (0.59 to 0.86)† | 0.78 (0.66 to 0.93)† | 0.98 (0.85 to 1.12) | 1.03 (0.90 to 1.18) |
| Human resource primacy climate | 0.70 (0.59 to 0.83)† | 0.77 (0.66 to 0.90)† | 0.88 (0.77 to 1.00) | 0.93 (0.82 to 1.05) |
| Innovative organisational climate | 0.79 (0.66 0.95)† | 0.82 (0.69 to 0.98)† | 0.95 (0.84 to 1.09) | 0.99 (0.86 to 1.13) |

OR (95% CI) for sickness absence from work (yes/no).
RR (95% CI) for number of sickness absence days among those with sick leave.
*Adjusted for time (T1 contrasted with T2), sex, age, education, sick leave prior to attack and direct exposure to the bomb explosion.
†Significant.
RR, rate ratio.

the workplace. Our findings indicate that a supportive psychosocial work environment can reduce the odds for sick leave by more than 20%. For the second part of the Hurdle-analysis, we observed no associations between a supportive psychosocial work environment and the duration of sickness absence. Further, we found that the proportion of women were higher among employees with sick leave when compared with those without sick leave.

Our first finding is in accordance with other studies (without focus on terrorism exposure) that find a protective effect of social support from the work environment, leading to a reduced risk of sick leave.[5–7] Especially, colleague support and appropriate supervision from the leader are important.[29 30] Moreover, previous findings show that high levels of social support at work are associated with reduced psychological distress, depression and increased well-being.[12 14–16] Evidence for the protective effect of social support on mental health and distress is relevant for our outcome on sick leave risk, as employment and sick leave frequency are markers of functional recovery from trauma.[2] Most people affected by disasters do not develop severe psychiatric disorders, though almost everyone with exposure to disaster trauma will experience distress for at least a brief period. A measure of sickness absence can capture individuals with subdiagnostic distress after terrorism.

Our second main finding showed that psychosocial support was not associated with a reduction in the duration of sickness absence, indicating that the protective role of a supportive psychosocial work environment was significant only when employees managed to stay at work. This can partly be explained by studies showing that the relationship between social support at work and sickness absence is bidirectional or reciprocal.[20 31] A study by Sieurin et al found that long-term absentees often reported that their absence negatively affected their sense of belonging to the social workgroup.[32] One

speculation is that sickness absence may affect the social relationships at work, and thereby adding to the challenges causing the sickness absence in the first place. Another factor could be that employees absent from work lose essential interaction with their work environment, and consequently, the positive effects of psychosocial support are reduced. Further, there might have been a 'threshold effect' in our sample, where employees with sick leave had reached a higher level of adverse health and distress, where coping strategies and support from the work environment no longer have an effect on the frequency of sickness absence days.

The utility of social resources depends on the survivors' ability to seek and receive support from their social and interpersonal resources in the aftermath of a terror attack.[16] Clearly, employees with sick leave had significantly more psychological strains and symptoms compared with those without sick leave, and they were slightly older with proportionally more women, which is in line with previous findings on sick leave risk with evidence for gender, older age and history of sickness as risk factors.[19 33–35] According to the social selection hypothesis, people struggling with post-traumatic stress symptoms such as withdrawal, depression and irritability may not seek necessary social support but instead diminish their interpersonal relations over time.[14] Further, employees experiencing psychological distress might evaluate the psychosocial work environment more negatively, affecting the subjective appraisals of leader and coworker support. This explanation can be related to a tendency in our results where leader support was not the most essential protective factor against sick leave for the employees after the terror attack. It could be that employees suffering from psychological distress after the attack perceived their immediate leader to be less supportive, as distress could have a negative effect on subjective appraisals of leader support.[36] However, the causation could as well be reversed, where employees

with a poor psychosocial work environment experience more psychological distress.

## Strengths and limitations

By means of a unique longitudinal design with registry data on sick leave, we were able to extend our knowledge of sick leave risk, and how it relates to the psychosocial work environment after terror. Existing research on terrorism and health effects are mostly focused on psychotraumatology and mental disorders such as PTSD, and sick leave may be a better objective measure of general functioning and health, especially consistently and routinely collected doctor certified data.[6 37]

The study has several limitations. First, the study population consisted of a majority of highly educated government officials and bureaucrats and our findings may not necessarily be generalised to other populations.[38] Recent results from the same cohort study found that the sick leave rates for ministerial employees were lower than in the general Norwegian population prior to, as well after, the bomb explosion.[3] Furthermore, it can be that the terror attack could have greater negative health effects in another sample, as the sick leave rates were higher, for example among Norwegian tourists exposed to the South-East Asian tsunami in 2004.[39] Second, we used data on doctor-certified sick leave. In Norway, the workers are entitled to be home from work up to three consecutive days, four times per year, without doctor certification. Further, if the employer had signed the Agreement for a More Inclusive Working Life (Inkluderende Arbeidsliv Agreement), the employee could report sick leave for eight consecutive calendar days without doctor certification, up to a total of 24 days per year, without limiting the number of times. We assume that many governmental employees were covered by the IA Agreement during the follow-up period. Therefore, the difference between registered and actual sick leave may be larger than anticipated, measured as incidence as well as days. Therefore, the overall incidence rate of sick leave would be higher if this was included in the official registry. Third, it should be noted that sickness absence is not only indicative of health problems but strongly affected by factors such as education, health behaviours, and sick pay insurance.[40] As such, sickness absence is not necessarily a precise measure of the health difficulties in the aftermath of a terrorist attack.

Fourth, the assessment of the psychosocial work environment was based on self-reported measures and may be influenced by social desirability, under-reporting, recall bias and other response bias, where subjective appraisals of the work environment are closely linked to personality traits and negative affect.[41] However, the QPSNordic has been validated as an effective instrument for measuring psychological and social factors at work and should be relatively insensitive to personality and emotions, for example, respondents were asked about how frequently a situation occurs instead of degrees of agreement or satisfaction.[22] Finally, we had no information on workplace social support prior to the terrorist attack, allowing a comparison of the effect of social support on sick leave before and after the terrorist attack. However, we had data on sick leave before the attack and were able to adjust for important confounders, such as previous sickness absence in the main analysis, a factor that could potentially influence later sick leave risk, and participant ratings of the psychosocial work environment.

## Implications

It seems reasonable to conclude that psychosocial support at work can reduce the sick leave risk after terror. The workplace should mobilise its social and supportive resources before employees become sick-listed, and before potentially new negative psychological and physical reactions have time to develop and at worse result in long-term sick leave. Especially as the psychosocial work factors appeared to have no major impact on the duration of sickness absence. Future research should investigate whether the associations are causal and whether these findings can be replicated in other populations over a longer time period. A more comprehensive approach that incorporates both intrinsic and extrinsic factors is needed to better capture and understand how individuals cope in the aftermath of terror and to identify the difficulties for return to work.

**Acknowledgements** We thank all the governmental employees who took part in this study.

**Contributors** TH designed and conducted the study and are responsible for the overall content. MTGD and AN analysed the data and MTGD drafted the paper. All authors, TH, MB, HKG, AN and MTGD, participated in project meetings where the analysis plan and data interpretation were discussed and where the article was critically revised.

**Competing interests** None declared.

**Patient consent for publication** Not applicable.

**Ethics approval** Informed consent was obtained from all participants, and the Regional Committee for Ethics in Medical Research approved the study (reference number: 2011/1577).

**Provenance and peer review** Not commissioned; externally peer reviewed.

**Data availability statement** Data are available on reasonable request. The datasets generated during and/or analyaed during the current study are not publicly available due to confidentiality agreements made with participants, but are available from the corresponding author on reasonable request.

**ORCID iDs**
Maria Teresa Grønning Dale http://orcid.org/0000-0002-0972-2996

Alexander Nissen http://orcid.org/0000-0003-2879-0457

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
