## [Reviewer comments · BMJ Open]

ARTICLE DETAILS

TITLE (PROVISIONAL)	Psychosocial work factors and sick leave risk after a terrorist bomb attack: a survey and registry-based longitudinal study of governmental employees in Norway.
AUTHORS	Dale, Maria Teresa; Nissen, Alexander; Berthelsen, Mona; Gjessing, Håkon; Heir, Trond

VERSION 1 – REVIEW

REVIEWER	Ingrid Mehlum The National Institute of Occupational Health, Dept. of Occupational Medicine and Epidemiology
REVIEW RETURNED	11-Jul-2021

GENERAL COMMENTS	The authors of this manuscript have explored prospectively the effects of the psychosocial work environment on sick leave risk among governmental employees after a workplace bomb attack. The manuscript is generally very interesting and well written; however, I have some comments. Follow-up: The participants were followed for 9 months after the surveys at T1 and T2, at 10 and 22 months, respectively, after the bomb attack, which should mean that the surveys were in May 2012 and May 2013, although, according to Fig. 1, the surveys seem to have taken place over 4 months, and it appears in the Results section that follow-up did not start in months 10 and 22, but in months 13 and 25, i.e., in August of those years, probably after most people had their summer vacation. This could preferably be described in more detail. The survey at T3 is mentioned but not used in this study, which is also a bit confusing for the reader. It might be better to leave it completely out of the manuscript. In the Results section, most results are reported over a period of 9 months follow-up, but some are reported over 12 months, which makes it difficult to relate these results to the main results. I would recommend using the same follow-up times for all results. Population: The study includes employees who were directly or indirectly exposed to the bomb. How were these two categories defined? It is stated that “All employees provided written consent” (p. 7, line 25) and in the following sentence that “3520 invited employees consented to participate”. Were these two different consents? When were the consent(s) collected? How many were invited to
--

	participate? Further, 1625 participants were eligible for the study (p. 7, line 33). Although the exclusion criteria may be seen in the Flow chart in Fig. 2, the inclusion and exclusion criteria could preferably be described in the Method section. It is not quite clear who the excluded group “Did not participate in web-based survey at any timepoint, n=1023” comprised. Were employees included if they participated at T1 and/or T2? If so, this should be stated explicitly, and also how much the populations at T1 and T2 overlapped. Outcome: The outcome is registry data on doctor-certified sick leave from the Norwegian Labour and Welfare Administration (NAV). It is stated that, “in Norway, the workers are entitled to be home from work up to three consecutive days, four times per year, without doctor certification” (p. 15, lines 22-25). However, if the employer had signed the Agreement for a More Inclusive Working Life (IA Agreement), the employee could report sick for 8 consecutive calendar days without doctor certification, up to a total of 24 days per year, without limiting the number of times. Governmental employees were most likely covered by the IA Agreement during the follow-up period. Therefore, the difference between registered and actual sick leave may be larger than anticipated, measured as incidence as well as days. The roles of the two variables psychological distress score and PCL-based PTSD are not quite clear. Are they outcomes? Could they be mediators? They are included in Table 1 and reported in the Results. They are even included in the summary of the main results in the first paragraph of the Discussion but still are not mentioned or defined in the Methods section. If they are to be included in this paper (or another paper), I suggest that their roles should be made clearer and examined more thoroughly in the analyses. Exposures and analyses: 5 psychosocial work environment variables (based on several items), representing the 2 dimensions “Social support/interactions” and “Organizational culture” were included in separate analyses. The scores for each item ranged from 1 to 5, however, the authors do not explain how these variables are included in the analyses, they used mean scores or cut-of values to define exposed/non-exposed. This is important in order to interpret the results. The authors concluded: “Our findings indicate that a supportive psychosocial work environment can reduce the odds for sick leave with more than 20 percent.” How much more supportive does the work environment need to be for the odds to be reduced by 20%? The effect of combinations of these psychosocial factors were not examined. Does more support from both superior and co-workers reduce the odds even more? Does the combination of more social support and a more positive organizational culture reduce the odds more? This could have been interesting to know. Some comments to different sections of the manuscript:
--	---

STRENGTHS AND LIMITATIONS:

Relatively high response rate: This is not explicitly reported in the manuscript.

METHODS:

Covariates (p. 9, line 42): The covariate “time” is very unspecific and should be defined. Is it calendar time, survey wave (T1 vs. T2) or what?

Analytic strategy: What does “at least one time point without missing values” (p. 9, bottom) mean? Without any missing values? Missing response to items of the psychosocial work exposures was allowed (p. 8, lines 13-15).

RESULTS:

P. 11, line 10: What are the following numbers: 259/1000 (421/1,625)?

Similarly for the numbers in line 15.

P. 11, line 18: “See Figure 1.” I cannot see that this figure explains what is described in the previous sentence(s).

I suggest moving the last part of the paragraph on p. 11 (lines 50-57) up, before line 30.

Description of the results of Table 1: Several of the numbers differ between the table and the text.

Similarly for Table 2, where one of the results must be wrong (the estimate and the lower confidence interval have the same value).

Table 1 includes the characteristics/distribution of the covariates and the outcome, as well as psychological distress and PTSD, but not the exposures, which should clearly be included in the table. See STROBE Statement, Item 14 (a).

The distributions of covariates (potential confounders) among employees with and without sick leave are compared and significance tested, shown in Table 1. However, significance testing of potential confounders is not recommended by leading epidemiologists. See, e.g., Hernberg S. Significance testing of potential confounders and other properties of study groups – misuse of statistics. *Scand J Work Environ Health* 1996;22(4):315-317.

In Table 2, P-values are included, in addition to 95% confidence intervals, which is not necessary.

According to the STROBE Statement (Item 16): (a) Give unadjusted estimates and, if applicable, confounder-adjusted estimates and their precision (eg, 95% confidence interval).

P. 12, line 20: “strongest estimates for HRP (OR=0.77...”.

The estimates for the three organizational climate variables are very similar and would all have been 0.8 if the results had been reported with one decimal only. The differences between them are probably not statistically significant.

DISCUSSION:

	P. 13, line 32: "...employment and sick leave frequency are markers of functional recovery from a trauma." Is this general knowledge? Could a reference be needed? P. 14, lines 35: Reference 34 studies low back pain. Is it relevant here? P. 14, lines 35-37: "...employees experiencing psychological distress might evaluate the psychosocial work environment more negatively." It could also be the other way around (reverse causation), that employees having poor psychosocial work environment (and report it being more negatively) experience psychological distress. P. 14, line 47: "distress could have a negative effect on subjective appraisals of leader support." And also for co-worker support? P. 15, lines 10-20: "It could be that the terror attack could have greater negative health effects in another sample, as the study sample was highly educated with flexible jobs, which is associated with lower incidence rate of sick leave". Odds ratio is a relative effect measure, and the sick leave incidence of the reference group (in the denominator) may have a large impact on the OR value. If another sample had lower education and higher sick leave incidence (higher value in the denominator), the result could easily be a smaller OR, not larger. 20: collected 42 months before the attack t REFERENCES The format of the references is not consistent. LANGUAGE The language is generally good, but there are a few issues. P. 3, line 20: "...collected FROM 42 months before the attack to 33 months after " (add "from") P. 6, line 46: "...a unique.." (not "an") P. 6, line 51: "Norwegian Labour" (not "Labor" if British English is used) P. 7, line 52: "involved IN" (add "in") STROBE Statement: The STROBE Statement has only been filled out for items 1-13, while statements for items 14-22 are missing. The pages referred to do not match the pages of the manuscript.
--	---

REVIEWER	Maria Melchior INSERM UMRS 1136 IPLESP
REVIEW RETURNED	12-Jul-2021

GENERAL COMMENTS	This study examines the role of psychosocial work factors as predictors of sickness absence among persons affected by a terror attack at work in Norway. The results show that high social support from coworkers is associated with reduced odds of sick leave after the terror attack, as is the social work climate. These results are original and potentially add to the literature, however I have some
---

	comments regarding aspects that should be better explained or modified.  1. The introduction does not mention non-work related factors in relation to sick leave after a terror attack, which should be described. 2. The duration of follow-up in the study should be clearly described and justified. Is it long enough to examine the research question at hand? 3. It is not clear in the methods section whether all participants were affected by the bombing. If that is the case, it would be good to specify if there are different levels of exposure or potentially negative consequences of the bombing among study participants. 4. How do levels of sickness absence in the study population compare with the general population? are they elevated? 5. Sickness absence is not only an indicator of health but also health behaviors, and very much related to the health insurance system in place. It would be useful to add a discussion of the extent to which it is a relevant indicator for the purposes of this study.
--	--

VERSION 1 – AUTHOR RESPONSE

Response to reviewers

Reviewer 1

1. Follow-up:

The participants were followed for 9 months after the surveys at T1 and T2, at 10 and 22 months, respectively, after the bomb attack, which should mean that the surveys were in May 2012 and May 2013, although, according to Fig. 1, the surveys seem to have taken place over 4 months, and it appears in the Results section that follow-up did not start in months 10 and 22, but in months 13 and 25, i.e., in August of those years, probably after most people had their summer vacation. This could preferably be described in more detail.

Author reply:

The timeline (figure 1) is correct. However, it is an accurate observation that the survey data was collected over a three to four months period, and that is why we used sick leave data from three quarters (9 months) and not a whole year to be sure that the exposure variables were collected before the outcome. To avoid unnecessary complexity, our research group has decided to report the collection time of the survey data within a consistent timeframe (one month) for all published articles. We have clarified this rationale in the manuscript, page 6.

2. The survey at T3 is mentioned but not used in this study, which is also a bit confusing for the reader. It might be better to leave it entirely out of the manuscript.

Author reply:

We agree and have excluded T3 from figure 1 and the whole manuscript. See new figure 1.

3. In the Results section, most results are reported over a period of 9 months follow-up, but some are reported over 12 months, which makes it difficult to relate these results to the main results. I would recommend using the same follow-up times for all results.

Author reply:

We have corrected the different periods in the Results from 13 to 21 month (T1) and 25-33 months (T2). See Page 10.

4. Population:

The study includes employees who were directly or indirectly exposed to the bomb. How were these two categories defined?

Author reply:

Proximity to the bomb explosion was assessed by asking employees where they were located during the explosion. Participants were given five alternative responses: 1) in the government district; 2) in downtown Oslo, but not in the government district; 3) in Oslo, but not downtown; 4) in Norway, but not

in Oslo; 5) abroad. Only employees in the government district when the bomb exploded were defined as directly exposed. This is now clarified in the Methods, Study population and data sources, page 6 and under Covariates page 9.

5. It is stated that "All employees provided written consent" (p. 7, line 25) and in the following sentence that "3520 invited employees consented to participate". Were these two different consents? When were the consent(s) collected? How many were invited to participate?

Author reply:

We have both use the term "written" and "informed consent" in this manuscript. The correct term is *informed consent*. This is now specified and elaborated on in the manuscript. Under METHODS, Study population and data sources, page 6. Further, we have included a brief description of which ministries (14 of 17) who consented to participate. Page 6.

6. Further, 1625 participants were eligible for the study (p. 7, line 33). Although the exclusion criteria may be seen in the Flow chart in Fig. 2, the inclusion and exclusion criteria could preferably be described in the Method section. It is not quite clear who the excluded group "Did not participate in web-based survey at any timepoint, n=1023" comprised. Were employees included if they participated at T1 and/or T2? If so, this should be stated explicitly, and also how much the populations at T1 and T2 overlapped.

Author reply:

We agree. 1023 employees did not participate in the survey at any timepoint (at T1 and T2), while 1061 participated at both T1 and T2. In this study population, we identified 1625 participants eligible for the study, all with relevant data on the psychosocial work environment at T1 **and/or** T2 and data for the 9- months following T1 and T2. This is now stated explicitly in the METHODS, Study population and data sources, page 7.

7. Outcome:

The outcome is registry data on doctor-certified sick leave from the Norwegian Labour and Welfare Administration (NAV). It is stated that, "in Norway, the workers are entitled to be home from work up to three consecutive days, four times per year, without doctor certification" (p. 15, lines 22-25). However, if the employer had signed the Agreement for a More Inclusive Working Life (IA Agreement), the employee could report sick for 8 consecutive calendar days without doctor certification, up to a total of 24 days per year, without limiting the number of times. Governmental employees were most likely covered by the IA Agreement during the follow-up period. Therefore, the difference between registered and actual sick leave may be larger than anticipated, measured as incidence as well as days.

Author reply:

Thank you for this important input. We have now added a paragraph regarding the IA Agreement in the manuscript.

8. The roles of the two variables psychological distress score and PCL-based PTSD are not quite clear. Are they outcomes? Could they be mediators? They are included in Table 1 and reported in the Results. They are even included in the summary of the main results in the first paragraph of the Discussion but still are not mentioned or defined in the Methods section. If they are to be included in this paper (or another paper), I suggest that their roles should be made clearer and examined more thoroughly in the analyses.

Author reply:

We agree that PTSD and SCL would be interesting to examine more thoroughly in the analysis. These two variables could both be mediators, moderators and even have a confounding effect. However, we think that we will "overadjust" for substantial variance in our outcome variable if we adjust for psychological distress (PTSD and PCL) in the analysis, as it is closely related to sick leave. Many of the complex mechanisms mentioned above can be approached and understood using a model inspired by the work of Wikmann et al. (2005), where psychological distress is closely related to our outcome. The central idea is that health can be conceptualized through the trilogy of illness, disease, and sickness (see Figure below). We have therefore decided to include these two in the descriptives (Table 1) as those

with sick leave have higher scores than those without sick leave both at T1 and T2. We think that doing a separate analysis for the different effects of PTSD and SCL is not within the main scope of this article, especially as they have previously been the main outcomes in other publications from our research group. We have therefore taken them out of the manuscript in the Discussion page 12 and from Table 1.

Hypothesised relation between environmental factors, trauma exposure and health problems by Wikman A, Marklund S, Alexanderson K. Illness, disease, and sickness absence: an empirical test of differences between concepts of ill health. *Journal of Epidemiology and Community Health* 2005;59(6):450-54.

9. Exposures and analyses:

5 psychosocial work environment variables (based on several items), representing the 2 dimensions "Social support/interactions" and "Organizational culture" were included in separate analyses. The scores for each item ranged from 1 to 5, however, the authors do not explain how these variables are included in the analyses, they used mean scores or cut-of values to define exposed/non-exposed. This is important in order to interpret the results.

Author reply:

Thank you for the observation. We used raw mean scores. This is specified in the Methods, Analytic strategy, page 10.

10. The authors concluded: "Our findings indicate that a supportive psychosocial work environment can reduce the odds for sick leave with more than 20 percent." How much more supportive does the work environment need to be for the odds to be reduced by 20%?

Author reply:

As we used raw mean scores, this means that if you increase the mean score with 1, the odds for sick leave is reduced by 20%.

11. The effect of combinations of these psychosocial factors were not examined. Does more support from both superior and coworkers reduce the odds even more? Does the combination of more social support and a more positive organizational culture reduce the odds more? This could have been interesting to know.

Author reply:

We agree that it would be interesting to look at both interactions and mediator effects between different combinations of these psychosocial factors. However, this would be a bit behind the scope of this article as we already have used each variable in five separate hurdle models.

12. Some comments to different sections of the manuscript: STRENGTHS AND LIMITATIONS:

Relatively high response rate: This is not explicitly reported in the manuscript.

Author reply:

The response rate was 56 % (n =1956). We have included this in the METHODS, Study population and data sources, page 7.

13. METHODS:

Covariates (p. 9, line 42): The covariate "time" is very unspecific and should be defined. Is it calendar time, survey wave (T1 vs. T2) or what?

Author reply:

Yes, time is the survey wave (T1 vs. T2). We have specified this in the METHODS, Covariates, page 7, and in table 2.

14. Analytic strategy: What does "at least one time point without missing values" (p. 9, bottom) mean? Without any missing values? Missing response to items of the psychosocial work exposures was allowed (p. 8, lines 13-15).

Author reply:

*Because we have used mixed effect models, the participants were included in the analysis as long as they had answered all variables (no missing values) included in the model for **at least one** survey wave (T1 and/or T2).*

When computing the mean sumscore for each psychosocial work exposure, we allowed a missing response to one item. These variables were later used in the mixed effect models. This is now clarified in the Methods, Analytic strategy, page 9 and under Psychosocial work exposures page 7.

15. RESULTS:

P. 11, line 10: What are the following numbers: 259/1000 (421/1,625)?

Similarly for the numbers in line 15.

Author reply:

They are confusing; this is now corrected to prevalence estimates in percent. See Results, Characteristics of the Study Population, page 11

16. P. 11, line 18: "See Figure 1." I cannot see that this figure explains what is described in the previous sentence(s).

Author reply:

This is now taken out; see Results, page 11.

17. I suggest moving the last part of the paragraph on p. 11 (lines 50-57) up, before line 30.

Author reply:

We agree; see Results page 11

18. Description of the results of Table 1: Several of the numbers differ between the table and the text.

Author reply:

We did the correction, see Results page 11

19. Similarly for Table 2, where one of the results must be wrong (the estimate and the lower confidence interval have the same value).

Author reply:

We did the correction in Table 2.

20. Table 1 includes the characteristics/distribution of the covariates and the outcome, as well as psychological distress and PTSD, but not the exposures, which should clearly be included in the table. See STROBE Statement, Item 14 (a).

Author reply:

Thank you for this input. We agree and have now included them in Table 1 and in the Results page 10.

21. The distributions of covariates (potential confounders) among employees with and without sick leave are compared and significance tested, shown in Table 1. However, significance testing of potential confounders is not recommended by leading epidemiologists. See, e.g., Hernberg S. Significance testing of potential confounders and other properties of study groups – misuse of statistics. *Scand J Work Environ Health* 1996;22(4):315-317.

Author reply:

Thank you for this article. We were not aware that this was common practice in epidemiology and have experienced that several recent publications report these estimates. However, we believe it is appropriate and meaningful to report these estimates and that it fits with the STROBE statement: "Before addressing the possible association between exposures (risk factors) and outcomes, authors should report relevant descriptive data. It may be possible and meaningful to present measures of association in the same table that presents the descriptive data." We, therefore, hope Table 1 is acceptable. (See JP Vandembroucke et al. (2007) *Strengthening the Reporting of Observational Studies in Epidemiology (STROBE): Explanation and Elaboration. Ann Intern Med.* 2007;147:W-163–W-194).

22. In Table 2, P-values are included, in addition to 95% confidence intervals, which is not necessary.

According to the STROBE Statement (Item 16): (a) Give unadjusted estimates and, if applicable, confounder-adjusted estimates and their precision (eg, 95% confidence interval).

Author reply:

We did the correction and have taken out the P-values from Table 2

23. P. 12, line20: "strongest estimates for HRP (OR=0.77...".
The estimates for the three organizational climate variables are very similar and would all have been 0.8 if the results had been reported with one decimal only. The differences between them are probably not statistically significant.

Author reply:

We agree and did the correction in the Results, *Psychosocial support at work and sick leave risk*, page 12, and in the Abstract page 2.

24. DISCUSSION: P. 13, line 32: "...employment and sick leave frequency are markers of functional recovery from a trauma."

Is this general knowledge? Could a reference be needed?

Author reply:

We agree. See Weiseth L, Heir T. *Workplace and organizational disasters: response and planning.* In: Ursano R, (Ed.), *Textbook of disaster psychiatry.* Cambridge: Cambridge University Press 2017:261-69.

25. P. 14, lines 35: Reference 34 studies low back pain. Is it relevant here?

Author reply:

Reference 34 is now taken out from our reference list.

26. P. 14, lines 35-37: "...employees experiencing psychological distress might evaluate the psychosocial work environment more negatively."

It could also be the other way around (reverse causation), that employees having poor psychosocial work environment (and report it being more negatively) experience psychological distress.

Author reply:

Yes, this perspective is now included. See Discussion, page 14.

27. P. 14, line 47: "distress could have a negative effect on subjective appraisals of leader support." And also for coworker support?

Author reply:

Yes, this is now included. See Discussion, page 14.

28. P. 15, lines 10-20: "It could be that the terror attack could have greater negative health effects in another sample, as the study sample was highly educated with flexible jobs, which is associated with lower incidence rate of sick leave".

Odds ratio is a relative effect measure, and the sick leave incidence of the reference group (in the denominator) may have a large impact on the OR value. If another sample had lower education and higher sick leave incidence (higher value in the denominator), the result could easily be a smaller OR, not larger. 20: collected 42 months before the attack t

Author reply:

We have now taken out this sentence. See Discussion page 15.

29. REFERENCES. The format of the references is not consistent.

Author reply:

We have tried to format the reference list.

30. LANGUAGE The language is generally good, but there are a few issues.

P. 3, line 20: "...collected FROM 42 months before the attack to 33 months after "(add "from")

P. 6, line 46: "...a unique.." (not "an") P. 6, line 51: "Norwegian Labour" (not "Labor" if British English is used) P. 7, line 52: "involved IN" (add "in")

Author reply:

We did all the corrections above.

31. STROBE Statement: The STROBE Statement has only been filled out for items 1-13, while statements for items 14-22 are missing. The pages referred to do not match the pages of the manuscript.

Author reply:

We have now included the relevant statements to each reporting item in the STROBE checklist.

Reviewer 2

1. The introduction does not mention non-work related factors in relation to sick leave after a terror attack, which should be described.

Author reply:

We have added a paragraph with this focus, see Introduction page 4.

2. The duration of follow-up in the study should be clearly described and justified. Is it long enough to examine the research question at hand?

Author reply:

Due to reviewer 1 we have now taken out the description of T3 from the manuscript as the rationale behind it was that since we do not use the data from this wave, it is also a bit confusing for the reader. We agree that it would be better to leave it entirely out of the manuscript. However, 34 months after the attack (T3), there was a significant governmental change and reorganization following the 2013 Norwegian parliamentary election. This may have affected the subjective ratings on the psychosocial work environment, with a major shift in leadership and organization culture. More specifically, the risk would be that data on perceived psychosocial work environment would refer to one work environment prior to restructuring and be used to predict sick leave in another work environment after restructuring. Therefore, we did not include this period (T3) in the analysis. We have added that future studies should investigate this research question over a more extended time period under Implications, page 16.

3. It is not clear in the methods section whether all participants were affected by the bombing. If that is the case, it would be good to specify if there are different levels of exposure or potentially negative consequences of the bombing among study participants.

Author reply:

Proximity to the bomb explosion was assessed by asking employees where they were located during the explosion. Participants were given five alternative responses: 1) in the government district; 2) in downtown Oslo, but not in the government district; 3) in Oslo, but not downtown; 4) in Norway, but not in Oslo; 5) abroad. Only employees in the government district when the bomb exploded were defined as directly exposed. This is now clarified in the Methods, Study population and data sources, page 6 and under Covariates page 9.

4. How do levels of sickness absence in the study population compare with the general population? are they elevated?

Author reply:

The sick leave rates for ministerial employees were lower than in the general Norwegian population prior to, as well as after, the bomb explosion. See publication from Hansen MB, Berthelsen M, Nissen A, et al. (2019) "Sick leave before and after a work-place targeted terror attack". International archives of occupational and environmental health.

We have included this in the Discussion, Strengths and limitations, page 15.

5. Sickness absence is not only an indicator of health but also health behaviors, and very much related to the health insurance system in place. It would be useful to add a discussion of the extent to which it is a relevant indicator for the purposes of this study.

Author reply:

We agree and have added this perspective in the limitations with a new reference (see M. Palme et al (2020) in the Discussion pages 15 and 16:

"It should be noted that sickness absence is not only indicative of health problems, but strongly affected by factors such as education, health behaviors and sick pay insurance. As such, sickness absence is not necessarily a precise measure of the health difficulties in the aftermath of a terrorist attack."